# Genetic Diversity of Nanmu (*Phoebe zhennan* S. Lee. et F. N. Wei) Breeding Population and Extraction of Core Collection Using nSSR, cpSSR and Phenotypic Markers

**Yan Zhu** [1], **Wenna An** [1], **Jian Peng** [2], **Jinwu Li** [2], **Yunjie Gu** [2,*], **Bo Jiang** [3], **Lianghua Chen** [1], **Peng Zhu** [1] **and Hanbo Yang** [1,*]

1  Forestry Ecological Engineering in the Upper Reaches of the Yangtze River Key Laboratory of Sichuan Province & National Forestry and Grassland Administration Key Laboratory of Forest Resources Conservation and Ecological Safety on the Upper Reaches of the Yangtze River & Rainy Area of West China Plantation Ecosystem Permanent Scientific Research Base, Institute of Ecology & Forestry, Sichuan Agricultural University, Chengdu 611130, China
2  Sichuan Key Laboratory of Ecological Restoration and Conservation for Forest and Wetland, Sichuan Academy of Forestry, Chengdu 610081, China
3  Du Fu Thatched Cottage Museum, Chengdu 610072, China
*  Correspondence: guyunjie123@gmail.com (Y.G.); yanghanbo6@sicau.edu.cn (H.Y.); Tel.: +86-182-8136-6168 (H.Y.)

**Abstract:** Genetic characterization is vital for tree germplasm utilization and conservation. Nanmu (*Phoebe zhennan* S. Lee. et F. N. Wei) is an extremely valuable tree species that can provide logs for many industrial products. This study aimed to assess the genetic diversity of a Nanmu breeding population using nine nSSR, five newly-developed cpSSR markers, and nine phenotypic traits, and extract a core collection. In general, the Na, Ne, and PIC for each nSSR/cpSSR were 2–37/2–3, 1.160–11.276/1.020–1.940, and 0.306–0.934/0.109–0.384, respectively. Fifteen chlorotype haplotypes were detected in 102 germplasms. The breeding population exhibited a relatively high level of genetic diversity for both nSSR (I = 1.768), cpSSR (I = 0.440, h = 0.286), and phenotypic traits ($H'$ = 1.98). Bayesian and cluster analysis clustered these germplasms into three groups. The germplasms revealed a high level of admixture between clusters, which indicated a relatively high level of gene exchange between germplasms. The F value (0.124) also showed a moderate genetic differentiation in the breeding population. A core collection consisting of 64 germplasms (62.7% of the whole germplasm) was extracted from phenotypic and molecular data, and the diversity parameters were not significantly different from those of the whole germplasm. Thereafter, a molecular identity was made up for each core germplasm. These results may contribute to germplasm management and conservation in the Nanmu breeding program, as well as genetics estimation and core collection extraction in other wood production rare species.

**Keywords:** *Phoebe zhennan*; genetic diversity; phenotypic; nSSR; cpSSR; core collection

## 1. Introduction

Successful breeding programs depend on abundant genetic diversity and the availability of genetic variation, which are the foundation of efficient selection and crop improvement [1]. The abundance of genetic variation is a prerequisite and guarantee of long-term genetic gains in tree breeding programs [2]. In tree breeding practice, the altering and reduction of genetic diversity could exist in tree breeding populations caused by the recurrent selection of superior accessions [3]. Consequently, genetic evaluation of a breeding population is a prerequisite in tree breeding programs [4]. However, relatively little research has been implemented on trees and little information on genetic diversity is known about tree breeding populations [4–6]. Abundant germplasm resources can provide a broad genetic foundation for breeding and genetic research [7]. Achieving this requires lots of energy and

time to investigate, collect and conserve as many germplasms as possible for a tree breeding program to increase the genetic diversity of the breeding population. On the other hand, the amount of resources presents a challenge for their effective conservation, management, and utilization [8,9]. Opportunely, core collection provides an effective method to resolve this issue, which, to the greatest extent, represents the genetic diversity of the totality of germplasms with minimum redundancy, and has been widely used in the utilization and conservation of plant genetic resources [10].

Phenotypic and genetic markers can be used for genetic diversity analysis and core collection extraction. Commercial phenotypic traits in wood plants that could be valuable goals in a tree breeding program include tree height (H), diameter at breast height (DBH), wood density, etc. Genetic diversity determination and core collection extraction using phenotypic traits is a relatively mature method that can reduce workload [11]. Yet, phenotypic traits are easily affected by environmental factors and do not accurately reflect differences at the DNA level [11,12]. Molecular markers, such as microsatellite markers (SSR), random amplified polymorphic DNA (RAPD) and restriction fragment length polymorphism (RFLP), which can reflect the polymorphism of DNA fragments and are not easily affected by the environment, have been widely adopted as powerful tools for germplasm characterization, cultivar identification, and diversity analysis in many plants [13–15]. However, the molecular markers only reflect DNA variation, which is not necessarily expressed in the phenotype [16]. The use of genotypic or molecular markers alone for genetic diversity and core collection extraction may not efficiently capture the whole genetic diversity of species [11,16]. Thus, combined phenotypic and molecular marker analysis can effectively reveal genetic diversity, prevent the loss of valuable germplasms in the progress of core collection extraction and promote the accuracy and reliability of core collection [17]. In tree species, determination of genetic diversity and core collection extraction have been constructed in several economic trees, such as Persian walnut (*Juglans regia* L.) [16], Carob tree (*Ceratonia siliqua*) [18], Pomegranate (*Punica granatum*) [19], and Olive (*Olea europaea*) [20]. However, core collection has been recently exploited in only a few timber species under the basis of genetic diversity determination, mainly concentrated in widely cultivated tree species such as Masson pine (*Pinus massoniana*) [1], Gympie messmate (*Eucalyptus cloeziana*) [4], Korean pine (*Pinus koraiensis*) [21], and Huangxinzimu (*Catalpa fargesii*) [11].

Nanmu (*Phoebe zhennan* S. Lee et F. N. Wei) belongs to the family of Lauraceae and genus of *Machilus* Nees and is mainly distributed in the subtropical evergreen broad-leaved forests (EBLFs) of China [22,23]. Nanmu is a major source of the well-known wood "Golden-thread nanmu", which is extremely valuable due to its durability, unique special fragrance, and attractive golden color [24]. Nanmu has a long history of practical use in coffins, palaces, and furniture since the pre-Qin Dynasty times (before 221 BC) [25]. As a highly economic and valuable wood, Nanmu is widely cultivated in China, especially in Sichuan province. However, there are few works on the genetic characteristic of this tree species in breeding programs, which greatly limits its breeding process and utilization. The chloroplast chromosome is non-recombinant, uniparentally inherited in many angiosperms and chloroplast microsatellite markers (cpSSR) are widely used in genetic and evolutionary studies of plants [26]. In recent years, using molecular markers derived from different genomes has shown to be a particularly efficient approach to the comprehensive analysis of the genetic diversity in numerous plants [27,28]. Therefore, in the present study, we examined the genetic diversity of *P. zhennan* germplasms. The objectives of this study were to (1) reveal the characteristics of genetic diversity at both a molecular level and a phenotypic level, (2) extract a core collection, and then to (3) provide a foundation for effectively utilizing the genetic resources of *P. zhennan*. Preliminary studies showed that low genetic diversity and isolation-by-distance has existed in Nanmu natural populations [22,24]. Therefore, the phenotypic and genetic information could be helpful for germplasm resource management and utilization of Nanmu breeding population in the breeding practice. To the best of our knowledge, this is the first report for the determination of the genetic diversity of Nanmu through joint molecular and phenotypic data analysis.

## 2. Materials and Methods

### 2.1. Germplasm Materials

The Nanmu (*Phoebe zhennan* S. Lee et. F. N. Wei) breeding population obtained 102 germplasms. The germplasms, which were collected from five provinces in China (65, 11, 17, 3, and 6 genotypes from Sichuan, Hubei, Hunan, Chongqing, and Guizhou Province, respectively, Table 1), were cultivated in Yuchan Mountain Nanmu Cultivation National Permanent Scientific Research Base (105.387529° E, 29.137715° N, 553 m altitude), Sichuan province, China in 2014. All of the 102 germplasms in the Nanmu breeding population were, among the main distributions of Nanmu, primary superior trees with excellent growth traits for height (H), diameter at breast height (DBH), and environmental adaptability.

**Table 1.** The origins of germplasms used in this study.

| Origin | Number of Germplasm | Germplasm Code | Latitude/° | Longitude/° |
|---|---|---|---|---|
| Sichuan province, China | 65 | ZNsc01-ZNsc65 | 30.8462 | 103.5600 |
| Hubei province, China | 11 | ZNhb01-ZNhb11 | 30.2669 | 108.9639 |
| Hunan province, China | 17 | ZNhn01-ZNhn17 | 29.3472 | 110.0844 |
| Chongqing province, China | 3 | ZNcq01-ZNcq03 | 29.3270 | 107.7538 |
| Guizhou province, China | 6 | ZNgz01-ZNgz06 | 27.1263 | 106.7335 |

### 2.2. DNA Isolation and SSRs Amplification

Fresh leaves from each individual were placed in a liquid nitrogen tank, transported to the laboratory, and frozen at −80 °C until DNA extraction. DNA was extracted using a Plant DNA isolation reagent (Takara Bio Inc., Dalian, China). The purity and quality of extracted DNA were evaluated by 0.8% agarose gel electrophoresis and determined using a NanoDrop 2000 spectrophotometer. Nine published nrSSR primer pairs [29,30], and 21 developed cpSSR primer pairs designed by SSRHunter1.3 (Nanjing Agricultural University, Nanjing, China) [31] basis on the Nanmu chloroplast sequence [32], were selected, and used to perform PCR analysis (Table S1). DNA amplification was carried out in a total volume of 25 μL containing 20 ng genomic DNA, 0.2 μM primers, 12.5 μL EmeraldAmp MAX HS PCR Master Mix (Takara Bio Inc., Dalian, China), and ddH₂O up to a total volume of 25 μL. The PCR thermal profile was the following: an initial step at 94 °C for 4 min, followed by 35 cycles at 94 °C for 30 s, 56–60 °C for 30 s and 72 °C for 1 min, and a final extension of 72 °C for 5 min. The amplified SSR loci labeled with fluorescent dyes were separated by capillary electrophoresis using an ABI 3500 automatic sequencer (Applied Biosystems, Foster City, CA, USA). The chromatograms were analyzed with the GeneMapper software version 3.2 (Applied Biosystems, Foster City, CA, USA).

### 2.3. Genetic Diversity Analyses

For nrSSR analysis, the GenAlEx 6.5 software (Rutgers University, New Brunswick, NJ, USA) [33] was used to calculate the number of different alleles (Na), the number of effective alleles (Ne), Shannon's information index (I), observed heterozygosity (Ho), expected heterozygosity (He), unbiased expect heterozygosity (uHe), and fixation index (F). In addition, we calculated the polymorphic information content (PIC) of each locus in CERVUS software version 3.0.7 (Montana State University, Bozeman, MT, USA) [34]. Based on Nei's genetic distance and the unweighted pair group method with arithmetic mean (UPGMA), a clustering was constructed using PowerMarker software version 3.25 (North Carolina State University, Raleigh, NC, USA) [35], and visualized using MEGA-X [36] and FigTree. The software STRUCTURE 2.3.4 (Stanford University, San Francisco, USA) [37] was used to analyze the genetic structure using ten independent runs for each *K* (1–10), and 100,000 generations of a burn-in period followed by 500,000 Markov Chain Monte Carlo (MCMC) iterations. A STRUCTURE HARVESTER [38] was used to determine the true value of *K*, which was inferred using an ad hoc quantity (Δ*K*). The average individual assignment probabilities over replicated runs were computed using CLUMPP version 1.1.2

(Stanford University, Stanford, CA, USA) [39]. The graphical STRUCTURE result was generated using DISTRUCT version 1.1 (Stanford University, San Francisco, USA) [40].

For cpSSR analysis, the GenAlex 6.5 software was also used to calculate the Na, Ne, I, haploid genetic diversity (h), and unbiased haploid genetic diversity (uh). A phylogenetic network tree was constructed using the median-joining model implemented in NETWORK 10 (https://www.fluxus-engineering.com/index.htm) [41] to investigate the relationships between chlorotypes.

### 2.4. Growth and Leaf Trait Measurement

The following traits were measured on living trees in November 2021: four growth traits, including tree height (H, m), diameter at breast height (DBH, cm), north-south crown diameter (PSN, m), and east-west crown diameter (PEW, m); one wood density trait, which was represented by Pilodyn value [42]; and four traits of leaf comprising leaf length (FI, cm), leaf width (LW, cm), petiole length (PI, cm), and the number of the secondary vein (SVN).

### 2.5. Phenotypic Traits Analysis

Descriptive statistics were carried out for whole individuals in the Nanmu breeding population. The phenotypic traits were divided into class 1 ($< \overline{X} - 2\delta$) to 10 ($\geq \overline{X} + 2\delta$) based on the mean value ($\overline{X}$) and standard deviation ($\delta$) of traits, with a $0.5\delta$ difference between each level. The morphological diversity for each phenotypic trait was calculated using the Shannon information index ($H'$) by the following formula [43]:

$$H' = - \sum_{n=1}^{n} Pi \times \ln(Pi)$$

where $Pi$ is the proportion of each trait in the germplasms, $\ln(Pi)$ is the natural logarithm of $Pi$. Furthermore, the correlation between phenotypic traits was calculated by Spearman's correlation analysis. The genetic dissimilarity component and agglomerative hierarchical clustering algorithm were analyzed and performed by Euclidean distance and Ward's method. Statistical analysis and plot drawing were performed by R 4.1.1.

### 2.6. Core Collection Extraction

Extraction of the core collection was performed based on analyses of phenotypic traits, nSSR, and cpSSR marker data, respectively, using the maximization strategy in the PowerCore v.1.0 software (National Institute of Agricultural Biotechnology, Suwon, Korea) [44]. First, three raw core collections were extracted by the data of phenotypic traits, nSSR, and cpSSR marker, respectively. Next, the three raw core collections were combined to form the final core collection to maintain the maximal genetic and phenotypic diversity. Analysis of genetic parameters and phenotypic traits between core collection and whole germplasms was determined by $t$-test ($\alpha = 0.05$). Additionally, principal component analysis (PCA) and principal coordinates analysis (PCoA) were used to verify the representation of the core collection. The $t$-test and PCA were performed by R 4.1.1, and PCoA was performed by GenAlEx 6.5. Finally, we established a molecular identity for the core collection using nSSR and cpSSR marker data. The band transport (A–Z and a–z) was used instead of PCR product size (bp), and '00' represented no band. The establishment steps for molecular identity refer to the method described by Zhong et al. (2021).

## 3. Results
### 3.1. Genetic Diversity of the Breeding Population
#### 3.1.1. Nuclear Genetic Diversity

A total of 153 alleles (Na) were detected using nine nSSR markers. As summarized in Table 2, the Na per locus ranged from 2 to 37 (mean, 17). The number of effective alleles (Ne) ranged from 1.160 to 16.015 (mean, 6.752), Shannon's information index (I) from 0.265 to 3.125 (mean, 1.768), observed heterozygosity (Ho) from 0.048 to 1.000, expected

heterozygosity (He) from 0.139 to 0.943, and fixation index (F) from −0.597 to 0.852. The mean unbiased expect heterozygosity (uHe) (0.659) and He (0.659) were higher than Ho (0.577), with a fixation index of 0.124. The polymorphic information content (PIC) indicated that the nine nSSR loci were relative highly polymorphic (mean value, 0.666), five microsatellites were shown to be highly polymorphic (PIC > 0.5), only one microsatellite (CL20730Contig1) exhibited low polymorphism (PIC = 0.128 < 0.25). The most polymorphic SSR locus (MN-g3) was 7.30-times higher than that of the least polymorphic locus (CL20730Contig1).

**Table 2.** Diversity statistics of the nSSR and cpSSR markers on 102 *P. zhennan* germplasms.

| Locus | Na | Ne | I | Ho | He | uHe | F | h | uh | PIC | NPA |
|---|---|---|---|---|---|---|---|---|---|---|---|
| **nSSR** | | | | | | | | | | | |
| ZiN-e8 | 10 | 1.476 | 0.761 | 0.048 | 0.325 | 0.325 | 0.852 | / | / | 0.306 | 8 |
| Unigene29601 | 8 | 2.894 | 1.354 | 0.228 | 0.658 | 0.658 | 0.651 | / | / | 0.617 | 4 |
| MN-e96 | 5 | 2.373 | 0.995 | 0.924 | 0.582 | 0.582 | −0.597 | / | / | 0.491 | 2 |
| MN-g30 | 33 | 11.276 | 2.885 | 1.000 | 0.916 | 0.916 | −0.097 | / | / | 0.906 | 17 |
| CL20730Contig1 | 2 | 1.160 | 0.265 | 0.149 | 0.139 | 0.139 | −0.080 | / | / | 0.128 | 0 |
| MN-g3 | 37 | 16.015 | 3.125 | 0.978 | 0.943 | 0.943 | −0.043 | / | / | 0.934 | 21 |
| MN-g5 | 34 | 14.387 | 3.029 | 0.278 | 0.936 | 0.936 | 0.701 | / | / | 0.926 | 21 |
| MN-g18 | 18 | 9.034 | 2.530 | 0.937 | 0.894 | 0.894 | −0.053 | / | / | 0.881 | 3 |
| CL4747Contig1 | 6 | 2.150 | 0.969 | 0.653 | 0.538 | 0.538 | −0.220 | / | / | 0.459 | 2 |
| Mean | 17 | 6.752 | 1.768 | 0.577 | 0.659 | 0.659 | 0.124 | / | / | 0.666 | 9 |
| **cpSSR** | | | | | | | | | | | |
| PZmk03 | 2 | 1.940 | 0.677 | / | / | / | / | 0.484 | 0.489 | 0.328 | / |
| PZmf07 | 2 | 1.147 | 0.250 | / | / | / | / | 0.128 | 0.129 | 0.320 | / |
| PZmf05 | 2 | 1.710 | 0.606 | / | / | / | / | 0.415 | 0.419 | 0.375 | / |
| PZmf06 | 3 | 1.625 | 0.609 | / | / | / | / | 0.385 | 0.388 | 0.384 | / |
| PZmf10 | 2 | 1.020 | 0.055 | / | / | / | / | 0.019 | 0.020 | 0.109 | / |
| Mean | 2 | 1.488 | 0.440 | / | / | / | / | 0.286 | 0.289 | 0.303 | / |

Na, number of different alleles; Ne, number of effective alleles; I, Shannon's information index; Ho, observed heterozygosity; He, expected heterozygosity; uHe, unbiased expected heterozygosity; F, fixation index; h, haploid genetic diversity; uh, unbiased haploid genetic diversity; PIC, mean polymorphic information content; NPA, number of private alleles in breeding population from five provinces.

### 3.1.2. cpSSR Polymorphism

The 21 designed cpSSR loci were firstly amplified in five individuals in the breeding population. However, eleven cpSSR loci were discarded from this study, because there were no amplification PCR productions detected. Five of the remaining ten loci were also discarded from this study due to the low-quality results reflected by the PIC value in ten individuals (Table S2). Finally, a total of five cpSSR primers were selected to be amplified in the *P. zhennan* breeding population. Five cpSSR loci used in this study generated a total of 11 Na across the breeding population (Table 2). Inside, the Na per locus ranged from 2 (PZmk03, PZmf07, PZmf05, and PZ mf10) to 3 (PZmf06), with an average of 2 Na per locus. The locus of PZmk03 showed the highest diversity and polymorphism (I = 0.677, h = 0.484, and uh = 0.489) in all five cpSSR loci, followed by PZmf05 and PZmf06.

### 3.2. Genetic Structure of Germplasms

A cluster analysis was performed to analyze the genetic relationships among the 102 germplasms in the Nanmu breeding population, and a dendrogram based on Nei's genetic distances and the unweighted pair group method with arithmetic mean is shown in Figure 1A. The cluster analysis divided the germplasms into three main groups, accounting for 18.63% (claret group), 23.53% (dark blue group), and 57.84% (yellow group) of all germplasms. Similar to the cluster analysis results, the results from STRUCTURE show that at K = 3, ΔK was optimal (Figure 1B), indicating that the most likely division of Nanmu germplasms included three sub-populations (Figure 1C). The sub-population 1 (light green) and sub-population 2 (orange) contained 39 and 49 germplasms, respectively. The main

germplasm in sub-population 1 and sub-population 2 were from Sichuan (35 germplasms in sub-population 1, 24 germplasms in sub-population 2), with a mixture of Hunan, Guizhou, Guizhou, and Chongqing. While the germplasms of Chongqing were only assigned to sub-population 2. The sub-population 3 (light yellow) contained 12 germplasm, which mainly came from Hubei (6/11), with a mixture of Sichuan and Hubei. Two germplasms (ZNsc13 and ZNsc46 from Sichuan) were not clustered in the above three sub-populations (Q value < 0.5 for every sub-population).

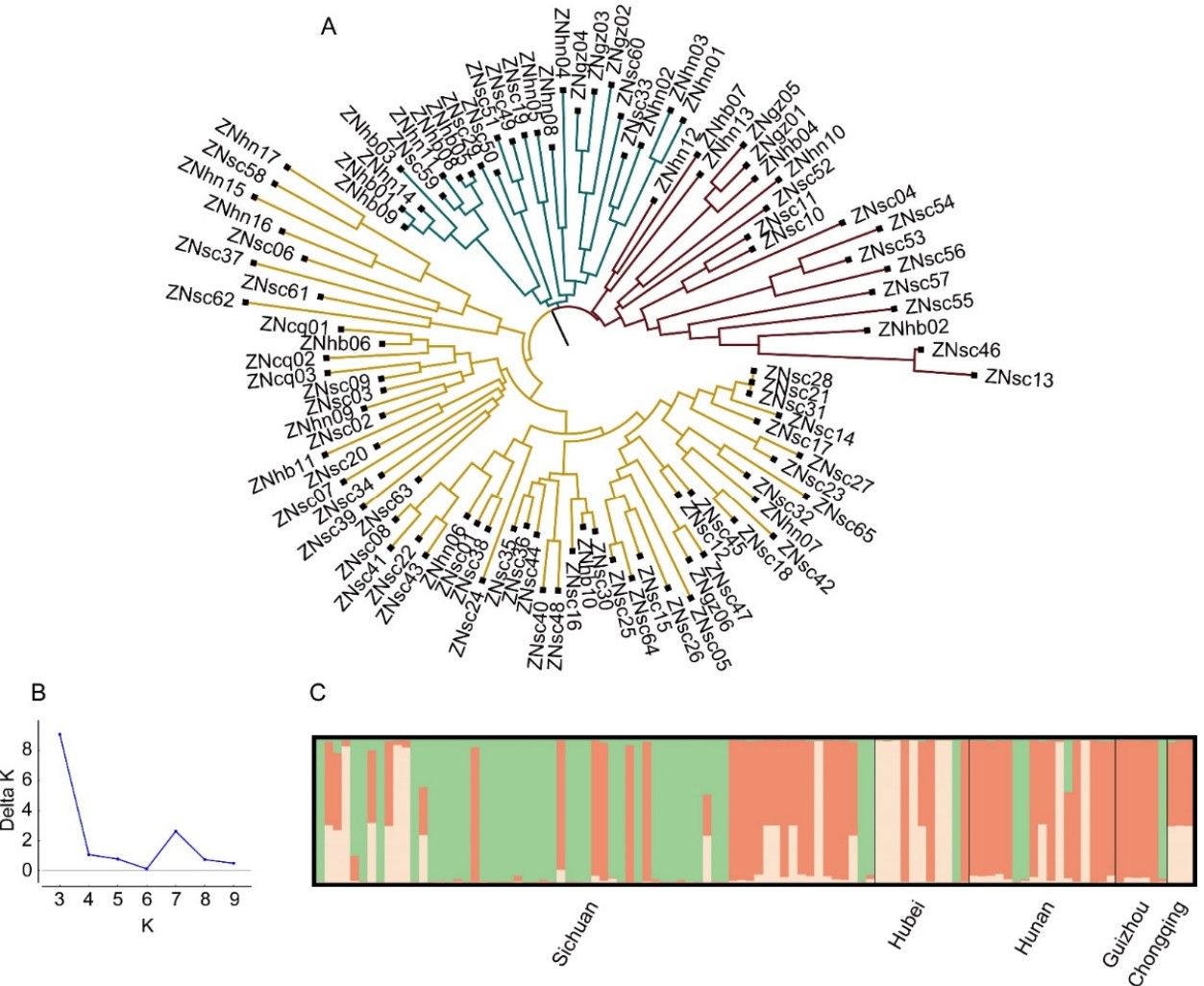

**Figure 1.** Phylogenetic and population structure analysis of 102 germplasms. (**A**) Phylogenetic tree of 102 germplasms based on genetic distance. The different colors represent three different groups (claret, dark blue and yellow group) of the whole germplasms which was divided by genetic distance among germplasms. (**B**) Delta K based on the rate of change of L(K) between successive K-values. (**C**) Population structure based on K = 3. The different colors represent three different sub-populations (light-green, orange, and light-yellow sub-populations) according to the Q value of STRUCTURE analysis.

### 3.3. Chloroplast Haplotypes Variation

A network approach was used to improve understanding of the genetic relationships between different chloroplast haplotypes (Figure 2). A total of 15 different chloroplast haplotypes in *P. zhennan* breeding populations were identified according to the results of the median-joining network (Table 3). Five (H1, H2, H9, H10, and H13) of the 15 chloroplast haplotypes were found in only one germplasm. H11 was the most frequent in 30/102 germplasms, followed by H3 with a frequency of 20/102 germplasms. There were

1–3 mutations between rare haplotypes (H1, H2, and H9, etc.) and advantage haplotypes (H3 and H11), indicated that these rare haplotypes would be the variance-type of H3 and H11.

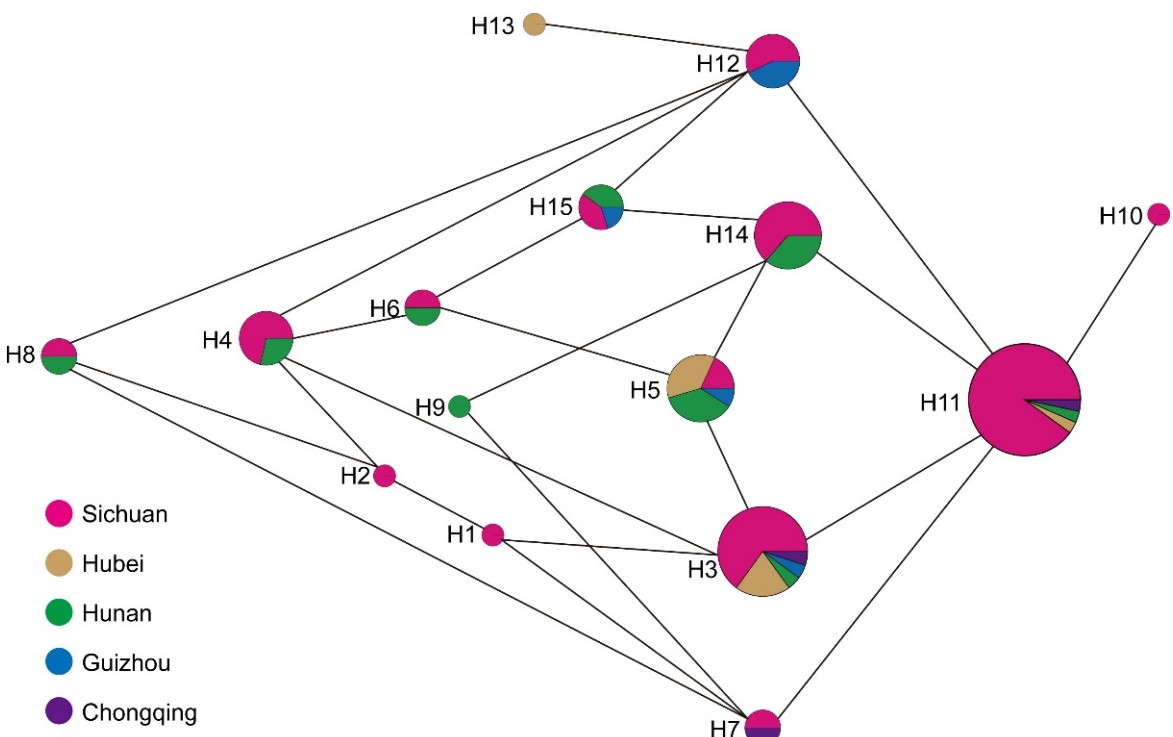

**Figure 2.** Median-joining haplotype network of 15 haplotypes identified in 102 germplasms. Circle areas are proportional to haplotype frequencies, each filled circle represents a different haplotype, the different colors represent the proportion of germplasms in every haplotype.

**Table 3.** Variation of haplotypes in five cpSSR loci.

| Haplotype | Count | PZmk03 | PZmf07 | PZmf05 | PZmf06 | PZmf10 |
|-----------|-------|--------|--------|--------|--------|--------|
| H1 | 1 | 179 | 267 | 238 | 204 | 222 |
| H2 | 1 | 179 | 267 | 238 | 205 | 222 |
| H3 | 20 | 179 | 268 | 238 | 204 | 222 |
| H4 | 7 | 179 | 268 | 238 | 205 | 222 |
| H5 | 11 | 179 | 268 | 239 | 204 | 222 |
| H6 | 2 | 179 | 268 | 239 | 205 | 222 |
| H7 | 2 | 180 | 267 | 238 | 204 | 222 |
| H8 | 2 | 180 | 267 | 238 | 205 | 222 |
| H9 | 1 | 180 | 267 | 239 | 204 | 222 |
| H10 | 1 | 180 | 268 | 238 | 203 | 222 |
| H11 | 30 | 180 | 268 | 238 | 204 | 222 |
| H12 | 7 | 180 | 268 | 238 | 205 | 222 |
| H13 | 1 | 180 | 268 | 238 | 205 | 223 |
| H14 | 11 | 180 | 268 | 239 | 204 | 222 |
| H15 | 5 | 180 | 268 | 239 | 205 | 222 |

Count means the number of germplasms with the same haplotype.

### 3.4. Phenotypic Diversity

The descriptive statistics of max, min, median, coefficient of variation (CV), and Shannon's information index (H′) are presented in Table S3. High diversity was found in phenotypic traits (H′ = 1.90–2.08, average value = 1.98). The order of the four germplasm populations was Sichuan > Hunan > Hubei > Guizhou according to the H′ value. The results of the Spearman's correlation analysis showed that tree height (H) showed a significant

positive correlation with the diameter at breast height (DBH), crown diameter (south-north crown diameter, PSN, and east-west crown diameter, PEW), and piloydn value (PV), but significant negative correlation with leaf width (LW) (Figure 3A). A significant positive correlation was observed between DBH, PSN, and PEW. There were also significant positive correlations between leaf length (FI) and LW, petiole length (PI), the number of secondary veins (SVN), LW and PI, SVN, PI, and SVN.

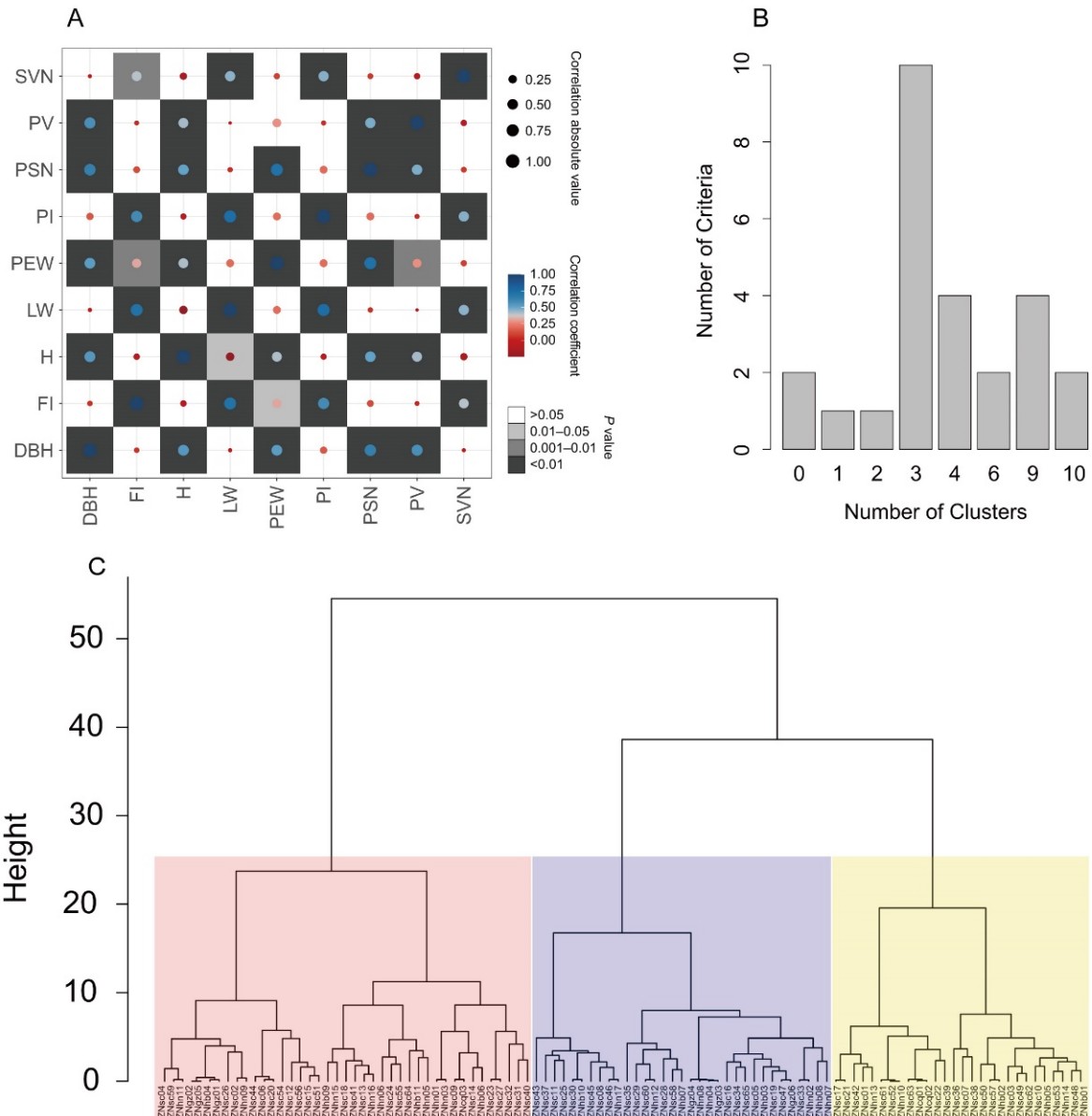

**Figure 3.** Correlation analysis among phenotypic traits (**A**) and ward cluster analysis (**B,C**) of *P. zhennan* breeding population. H, height; DBH, diameter at breast height; PSN, south-north crown diameter; PEW, east-west crown diameter; PV, piloydn value; FI, leaf length; LW, leaf width; PI, petiole length; SVN, the number of secondary veins; abs_cor means correlation absolute value; and value means correlation coefficient. The different colors (red, blue, and yellow) in the dendrogram represent three different groups which were divided by Euclidean distance.

According to the k-mean partitions comparison analysis based on the criteria, k = 3 was selected as the best k (Figure 3B). Consistent with the results of cluster and structure analysis by SSR markers, the cluster analysis of assayed germplasms, using the Ward's

method and Euclidean distance, revealed three main groups. The first (red), second (purple), and third (yellow) groups consisted of 41, 33, and 28 germplasms, respectively (Figure 3C).

### 3.5. Extraction of a Core Collection

Sixty four out of 102 germplasms were selected in the core collection using phenotypic traits, nSSR, and cpSSR data through PowerCore calculation, reducing the population size to 62.7% for the whole germplasms (Figure 4A, Table S4). The formed core collection consisted of the germplasms from Sichuan (38 genotypes, 58.5%), Hubei (5 genotypes, 45.5%), Hunan (13 genotypes, 76.5%), Guizhou (5 genotypes, 83.3%), and Chongqing (3 genotypes, 100.0%), respectively. In comparison with the whole germplasms, the final core collection showed 100.0% of the variation of Na, 122.0% and 105.7% of Ne and 106.0% and 109.9% of I values in nSSR and cpSSR, respectively. As shown in Table 4, there were no significant differences in genetic parameters and phenotypic traits between the final core collection and the whole germplasm. To further validate the core collection, PCoA and PCA of core collection and the whole germplasm were composed according to the molecular and phenotypic data, respectively (Figure 4B,C). The core collection showed similar cluster patterns with the whole germplasms. These results support the validity of the method for extracting the core collection and further suggest that the core collection effectively represents the whole germplasm. Finally, the molecular identity of each core collection was constructed according to the PIC value of SSR markers (Table S4).

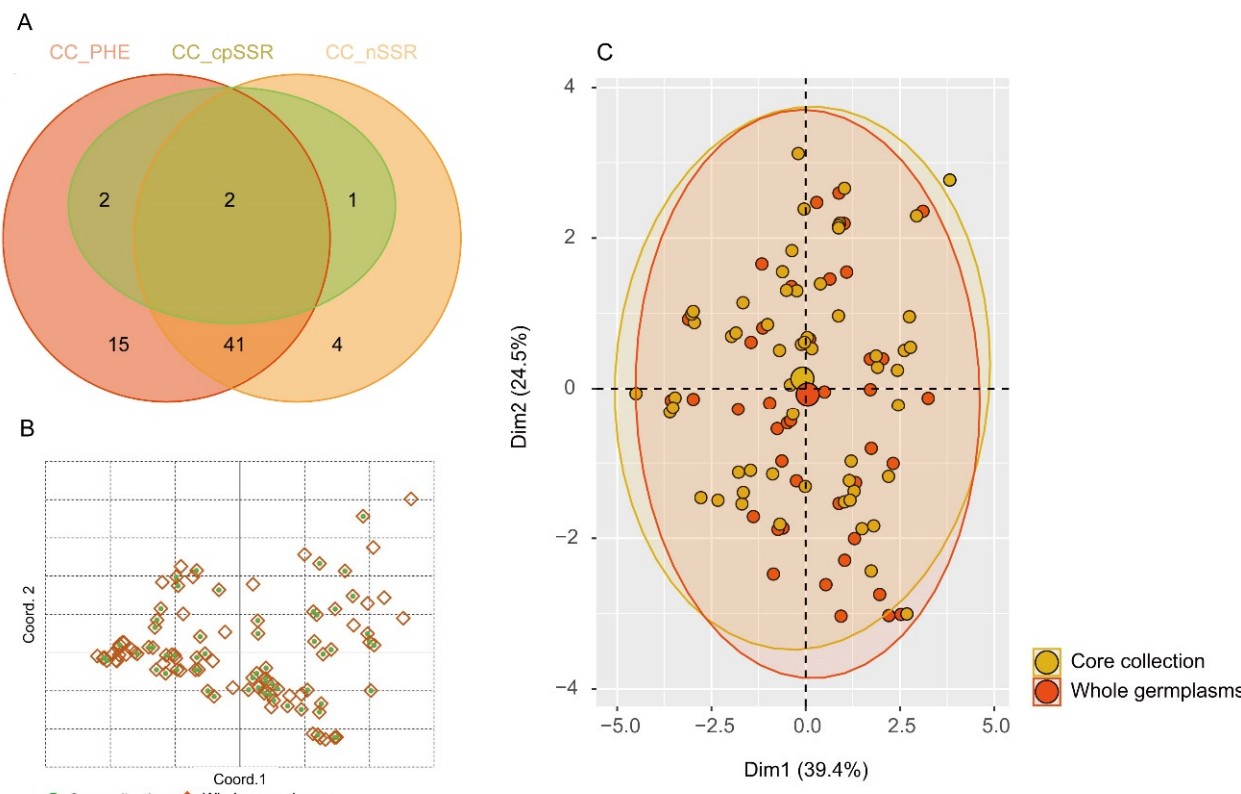

**Figure 4.** The extraction and representative verification of core collection with the whole germplasms. (**A**) Venn diagrams of phenotypic and SSR data building the core collection, CC_cpSSR, core collection extracted through cpSSR marker data, CC_nSSR, core collection extracted through nSSR marker data, CC_PHE, core collection extracted through phenotypic traits data. (**B**) Principal coordinates analysis (PCoA) plot of the core collection and whole collection. (**C**) Principal component analysis (PCA) plot of the core collection and whole collection.

**Table 4.** Comparison of phenotypic trait and genetic diversity between the core and raw locust sets.

| Traits | Core Collection | Whole Germplasms | Retention Rate/% | t Value | p Value |
|---|---|---|---|---|---|
| Phenotypic traits | | | | | |
| H | 5.1 | 5.3 | 96.3 | 0.532 | 0.597 |
| DBH | 5.3 | 5.5 | 95.7 | 1.023 | 0.311 |
| PSN | 2.3 | 2.4 | 96.5 | 0.533 | 0.596 |
| PEW | 2.4 | 2.4 | 97.3 | 0.494 | 0.623 |
| PV | 17.0 | 17.2 | 98.8 | 0.784 | 0.437 |
| FI | 8.9 | 8.7 | 101.4 | −0.204 | 0.839 |
| LW | 2.5 | 2.5 | 100.6 | −0.192 | 0.848 |
| PI | 0.8 | 0.8 | 101.4 | −0.212 | 0.833 |
| SVN | 21.5 | 21.5 | 99.9 | 0.450 | 0.655 |
| nSSR | | | | | |
| Na | 17 | 17 | 100.0 | 0.000 | 1.000 |
| Ne | 8.238 | 6.752 | 122.0 | 0.455 | 0.656 |
| I | 1.874 | 1.768 | 106.0 | 0.199 | 0.845 |
| Ho | 0.588 | 0.577 | 101.9 | 0.060 | 0.953 |
| He | 0.681 | 0.655 | 103.9 | 0.191 | 0.851 |
| uHe | 0.686 | 0.659 | 104.2 | 0.206 | 0.840 |
| cpSSR | | | | | |
| Na | 2.200 | 2.200 | 100.0 | 0.000 | 1.000 |
| Ne | 1.572 | 1.488 | 105.7 | 0.322 | 0.756 |
| I | 0.481 | 0.437 | 109.9 | 0.246 | 0.812 |
| h | 0.318 | 0.285 | 111.5 | 0.249 | 0.810 |
| uh | 0.323 | 0.288 | 112.2 | 0.263 | 0.800 |

H, height/m; DBH, diameter at breast height/cm; PSN, crown diameter from south to north/m; PEW, crown diameter from east to west/m; PV, piloydin value; FI, leaf length/cm; LW, leaf width/cm; PI, petiole length/cm; SVN, the number of secondary veins; Na, number of different alleles; N, number of effective alleles; I, Shannon's information index; Ho, observed heterozygosity; He, expected heterozygosity; h-haploid genetic diversity; uh, unbiased haploid genetic diversity.

## 4. Discussion

Understanding genetic diversity is the foundation and prerequisite for breeding and conservation of plant genetic resources [4,45]. The breeding progress of Nanmu (*Phoebe zhennan* S. Lee et. F. N. Wei) has been slow, in part because of slow growth, and the wood maturity stage is 60–80 years [46,47]. Furthermore, little is known about the genetic background of Nanmu making it difficult to select elite parents for further breeding. The present study comprehensively characterized the phenotypic and molecular genetic diversity of this breeding population. Our results will not only deepen our understanding of the genetic diversity but also facilitate its rational utilization in the Nanmu breeding population. The 102 Nanmu germplasms in this study showed relatively high genetic diversity based on phenotypic traits, nSSR, and cpSSR marker analysis. This outcome is contrary to that of Gao et al. (2016) who found low genetic diversity with four Nanmu natural populations at AFLP markers. This could be attributed to the SSR markers used in this study, which could be more polymorphic than AFLP markers, and reflect the wide genetic diversity in the Nanmu breeding population. Moreover, the breeding population showed relatively higher genetic diversity (I = 0.938) than that in natural populations of Nanmu (I = 0.0758–0.612), and elite genotype (I = 0.391) at ten microsatellites of Minnan (*Phoebe bournei*) [48]. The level of genetic differentiation (F = 0.124) observed in the Nanmu breeding population is lower than that observed in natural populations [22], indicating the significant influence of breeding on the genetic divergence of Nanmu populations. There is generally moderate genetic differentiation in breeding population, and migration is more important than genetic drift in our breeding population [49]. The chloroplast genome with low speed mutation plays a unique role in the genetic studies of populations with a high level of admixture or mutation [50]. Eleven alleles, (PIC = 0.303) detected in 102 germplasms from five cpSSR loci, indicated inferior polymorphism and conservation of chlorotypes genome in *P. zhennan*. The chlorotype of each germplasm was determined based on polymorphisms

at five cpSSR loci, which allowed the differentiation of all described chlorotypes. The chlorotype network indicated that H3 and H11 were the common ancestor to which several chloroplast haplotypes were related (Figure 2). In addition, H1, H2, H9, H10, and H13 were found only in one germplasm, possibly because it came from a site in which there may have been little gene flow (H1, H2, H10-Sichuan), or a limited sample was analyzed in this study (H9-Hunan, H13-Hubei) [27]. cpSSR analysis can allow evaluation of lineages of hypothetical maternal origin [27]. Our results suggest that Sichuan province would be the center of origin of *P. zhennan*, carrying the dominant chloroplast haplotypes H3 and H11. Our study found that Shannon's information index was much higher ($H' = 1.98$) when calculated using phenotypic data in the breeding population than when using nSSR ($I = 1.297$) and cpSSR molecular data ($I = 0.440$). The breeding population used in this study was extracted from the base population (natural population) based on phenotypic trait data, some genetically similar accessions were removed from the breeding population. Therefore, genetic diversity was comparatively higher when examined at the phenotypic level. Similar results have also been reported in sesame (*Sesamum indicum*) [7]. This is one of the reasons why the following core collection was extracted using a combination of phenotypic and molecular data, to prevent the loss of important economic phenotypes. There existed a significant positive relationship between Pilodyn value (PV) and height (H), diameter at breast height (DBH), south-north crown diameter (PSN), and east-west crown diameter (PEW). Accordingly, wood density and growth traits should be considered separately when the superior genotype should be selected for breeding because the wood density had a negative correlation with PV. On the whole, a potential genetic gain would be obtained in the following breeding practices because of the relatively high level of phenotypic and molecular genetic diversity in Nanmu breeding population.

Both STRUCTURE and cluster analyses contribute to a better comprehension of the moderate genetic differentiation in Nanmu breeding population [18]. STRUCTURE and Phylogenetic analyses, based on molecular and phenotypic traits, tend to a high level of admixture between clusters, indicated that introgression would exist in the breeding population through the exchange of seeds or seedlings between provinces. As for genetic structure and genetic distance, Xiao et al. (2020) and Gao et al. (2016) divided 12 Nanmu populations and 92 accessions into two or three groups, similar to the results of the present study. This may contribute to the weak genetic differentiation between five populations. Similar results have also been reported for Korean pine (*Pinus koraiensis*) [21] and Gympie messmate (*Eucalyptus cloeziana*) [4]. This study assessed genetic diversity in the three groups of the Nanmu breeding population both at the phenotypic and molecular levels, but the results are inconsistent. It is possible, therefore, that molecular measures of genetic diversity have a very limited ability to predict quantitative genetic variability [51]. Hence, combination of phenotypic and molecular analyses in genetic diversity assessments of Nanmu, or in other tree breeding population is very necessary.

Previous studies have noted the importance of core collection extraction in germplasm protection and utilization, especially in perennial tree plants. Up to now, core collection extraction has been reported in only a few timber tree species, for instance, Masson pine (*Pinus massoniana*) [1], Chinese fir (*Cunninghamia lanceolata*) [52], and Gympie messmate (*Eucalyptus cloeziana*) [4]. It is clear that molecular markers are more useful when used together with morphological markers [16]. To further reduce the duplication of some germplasms in the Nanmu breeding population, core collection (58.8%, 60/102) has been developed and evaluated for Nanmu germplasms through phenotypic, nSSR, and cpSSR marker data. The results of our study confirm the feasibility of core collection extraction in precious timber tree species. The core collection represented 62.7% of the whole collection, which is higher than the range of 5%–10% [53], as well as the values reported in other wood production trees, e.g., Masson pine (34.2%), Chinese fir (42.9%), and Gympie messmate (35%) [1,4,52]. Furthest retention of genetic diversity of the whole germplasm should be a priority when guiding the determination of core collection extraction under any circumstances [54], accordingly, we did not aim for a low rate of germplasm retention.

The results also indicate abundant phenotypic and molecular diversity of germplasms in our *P. zhennan* breeding population. This review is also corroborated by previous work in other plants [4,9]. In comparison with the whole collection, the final retention rate of the core collection was 95.7%–122.0% of all parameters of phenotypic traits, nSSR and cpSSR markers. The results of the PCoA and PCA also show a uniform distribution of core collection in the whole germplasm. These results indicate that the core collection could represent the genetic diversity of 102 germplasms in Nanmu breeding population. Furthermore, the molecular identity of the core collection will be useful for breeding management strategies. Moreover, core collection extraction should be a dynamic process. New germplasms with superior phenotypes and rich genetic variation can be complemented periodically [55]. Further, advanced marker technologies, such as genome-wide nucleotide polymorphism (SNP) can greatly facilitate the depth and accuracy of genome-wide variation [56]. Further breeding work may help early evaluation of the economic traits of Nanmu germplasms with genomic selection that shortens the evaluation cycle based on the significant and close correlation between phenotypic traits and genome-wide variation [57].

## 5. Conclusions

The present study revealed the relatively high genetic diversity and high level of gene exchange in the breeding population of Nanmu (*Phoebe zhennan*) based on nine nSSR, five new cpSSR, and nine phenotypic markers. A high level of admixture existed between clusters by Bayesian and cluster analysis, while only a part of the collection showed significant genetic variability, which will facilitate the breeding process. The phenotypic variability was not always consistent with molecular variability. A core collection was defined at a percentage of 62.7% of the whole germplasms, which effectively represented the whole germplasm. Knowledge of genetic diversity provided a basis for germplasm management and utilization in Nanmu, and core collection establishment for other valuable timber tree plants as well.

**Supplementary Materials:** The following supporting information can be downloaded at: https: //www.mdpi.com/article/10.3390/f13081320/s1, Table S1. Characteristics of 29 original cpSSR primers. Table S2. PIC test of 11 designed cpSSR primer pairs in ten germplasms. Table S3. Descriptive statistics of the phenotypic traits in Phoebe zhennan breeding population. Table S4. Members and molecular identity cards of the core collection.

**Author Contributions:** H.Y. and Y.G. conceived and designed the project. Y.Z., W.A. and J.P. collected the phenotypic data. H.Y., Y.Z., J.L. and B.J. performed molecular lab work and scored the markers. Y.Z., L.C. and P.Z. analyzed the data. Y.Z. and H.Y. wrote the original manuscript. Y.G., L.C. and P.Z. helped to improve the manuscript. All authors have read and agreed to the published version of the manuscript.

**Funding:** This research was financially supported by the Research Funds of Key Research and Development Project of Sichuan Province (2021YFYZ0032), Natural Science Foundation of Sichuan Province (2022NSFSC1062), National Undergraduate Training Program for Innovation and Entrepreneurship (202110626009), Cultivation of Scientific Research Interest Project for Undergraduate of Sichuan Agricultural University (2022311, 2022194) and Forest Ecosystem Improvement in the Upper Reaches of Yangtze River Basin Program (510201202038467).

**Conflicts of Interest:** The authors declare no conflict of interest.

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
