# Peer review of "Genetic Diversity of Nanmu (Phoebe zhennan S. Lee. et F. N. Wei) Breeding Population and Extraction of Core Collection Using nSSR, cpSSR and Phenotypic Markers"

_forests, doi:10.3390/f13081320_

Round 1
Reviewer 1 Report
About the paper, with my known in relation to genetic diversity, I think this a complete article because includes both genetic and morphologic/phenotypic (with the main characters of wood tree) markers. I think that the results are consistent because nSSR are more polymorphic that chloroplast SSR, where DNA are more conserved. The number of SSR, nine, I think it is enough for such study. And they found an high polymorphisms for some of them.
I have some doubts with the use of cpSSR because I have not work with them. I saw small differences between the alleles, only one pair of bases. I think that if they include this information it is because they can detect very well this difference. So, maybe, it is speculative to talk about diversification places with this information, with cpSSR, with this little differences, that are not very clear.
Also, it is a traditional study, so it is no very new, but necessary. How it is a species with few studies, it is interesting. And Mammu belong an important botany family that include important wild and cultivated species as laurel or avocado.
I think the first dendrogram with SSR should be bigger, it looks little.
I think you must check the reference correspondence, P. eg, number 38
Author Response
Dear Professor:
Thank you for your letter and for the reviewers’ comments concerning our manuscript entitled “Genetic diversity of Nanmu (Phoebe zhennan S. Lee. et F. N. Wei) breeding population and extraction of core collection using cpSSR, nSSR and phenotypic markers” (Manuscript ID: forests-1805236). Those comments are all valuable and very helpful for revising and improving our paper, as well as the important guiding significance to our researches. We have studied comments carefully and have made correction which we hope meet with approval. Revised portion are marked in red in the paper. And we also upload the word file of the point-by-point response in the attachment. The main corrections in the paper and the responds to yours comments are as following:
- I have some doubts with the use of cpSSR because I have not work with them. I saw small differences between the alleles, only one pair of bases. I think that if they include this information it is because they can detect very well this difference. So, maybe, it is speculative to talk about diversification places with this information, with cpSSR, with this little differences, that are not very clear.
Response to comment: The chloroplast chromosome is non-recombinant, uniparentally inherited in many angiosperms, and effectively haploid (Phumichai et al, 2015). Chloroplast is characterized by haploidy and with rare exception, a lack of recombination and uniparental inheritance (Birky, 1995). Chloroplast microsatellite (cpSSR) markers are widely used in population genetics and evolutionary studies of plants (Provan et al, 2001). One pair of bases was detected by cpSSR because the chloroplast microsatellite marker is characterized by haploidy in plants. Similar with previous genetic studies with cpSSR, the alleles of cpSSR were commonly detected from 2 to 4 (Phumichai et al, 2015; Lee et al, 2019). The frequency of haplotypes according to the alleles of cpSSR were used to determine the level of genetic divergence. In our study, five cpSSR markers were used to explore the haploid genetic diversity in Nanmu breeding population. Increasing the number of markers of cpSSR is an effective shortcut to increase the accuracy of the results of haploid genetic diversity. As reviewer pointed out that the little differences of cpSSR markers would made the results no very clear, whereas, we explored a relatively high haplotypes (15) and haploid genetic diversity (0.019-0.484, with the mean of 0.286) of those five cpSSR markers in Nanmu breeding population. The new cpSSR markers were first applicated in genetic diversity analysis of Nanmu, and the cpSSR markers provided a reference for other population genetics of Phoebe plants. The results of cpSSR genetic analysis also could provide an important part and supplementary for the genetic diversity analysis of Nanmu breeding population with nSSR genetic diversity.
Reference
Phumichai C, Phumichai T, Wongkaew A. Novel chloroplast microsatellite (cpSSR) markers for genetic diversity assessment of cultivated and wild hevea rubber. Plant Molecular Reporter, 2015, 33 (5): 1486-1498.
Birky CW. Uniparental inheritance of mitochondrial and chloroplast genes: mechanisms and evolution. Proceedings of the National Academy of Science, 1995, 92, 11331-11338.
Provan J, Powell W, Hollingsworth PM. Chloroplast microsatellites: new tools for studies in plant ecology and evolution. Trends Ecology Evolution, 2001, 16: 142-147.
Lee KJ, Lee GA, Lee JR, Sebastin R, Shin MJ, Cho GT, Hyun DY. Genetic diversity of sweet potato (Ipomoea batatas L. Lam) germplasm collected worldwide using chloroplast SSR markers. Agronomy, 2019, 9: 752.
- I think the first dendrogram with SSR should be bigger, it looks little.
Response to comment: As you suggested that we have adjusted the size and proportion of dendrogram in Figure 1. Please review the revised Figure 1
- I think you must check the reference correspondence, P. eg, number 38.
Response to comment: Considering the yours suggestion, we have checked and revised the reference correspondence in the text of manuscript. Please review line 139, 142.
Special thanks to you for your good comments.
We appreciate for your warm work earnestly, and hope that the correction will meet with approval.
Once again, thank you very much for your comments and suggestions.

Reviewer 2 Report
The study aimed to assess the genetic and phenotypic diversity of a Phoebe zhennan breeding population and extract a core collection. Nanmu has a wide range of cultivation in China. However, there are few works on the genetic characteristic of this species, which greatly limits its breeding process and utilization. The Authors, using nSSR, cpSSR markers, and phenotypic traits, fill the gap in knowledge on this subject. The strength of the work is that the authors show that only a part of the collection shows significant genetic variability, which will facilitate the breeding process, and phenotypic variability does not always go hand in hand with molecular variability. Dear Authors! Why didn't you write it in the conclusion?! This phenomenon has already been observed in other wild species, but the authors have proved it with their research. This is knowledge that every researcher should use in their biodiversity research.
I have some comments on the manuscript.
General comments:
-
The authors misuse the term "germplasm". This term is very imprecise, which sometimes makes it difficult to understand the content. Sometimes it is better to use: plant, individuals, genotype, genome, DNA, cpDNA, etc. For example "„... fresh leaves from each germplasm...” substitute „... fresh leaves from each individuales/plants...”
-
Coloring of figures:
The description of the colors should be included not only in the text but also in the legend under the figures.
- Figure 2 please enter brighter and more contrasting colors and in Figure 3 more contrasting. This will facilitate the analysis. -
Please add the table with the tested material: 1. column: name of the Provinces in China; 2. column: the designations of the test subjects (eg ZNcs1 - ZNcs ...); 3. column: the geographical coordinates of the Provinces. (Don't forget to change the order of the tables in the text!! :-)). This will make it easier for the reader to analyze Figure 1A, 3C and Table S4.
-
Please unify the order of the subchapters in the Methods and Results sections: Molecular research first, then phenotypic, or vice versa. Include this change in the title of your publication.
-
Please indicate more clearly the research objectives in the Introduction and the sections: Conclusions and Author Contributions .
Detailed comments:
-
Chapter: Materials and Methods - DNA Isolation. "Genomic DNA extractin" is not an appropriate term. "Genomic" is usually used to refer to nuclear DNA, and you have also studied cpDNA. I suggest you write "DNA extraction"
-
Figure 4B. is indistinct.
-
Table S1 and S2; First column: not "Marker" but "Locus"
-
Table S4: „Molecular ID” better: „Molecular identity". The „ID” is usually restricted to official databases.
-
The names of Chinese provinces should consistently be spelled without quotation marks.
I have indicated other comments in the text.

Author Response
Dear Professor:
Thank you for your letter and comments concerning our manuscript entitled “Genetic diversity of Nanmu (Phoebe zhennan S. Lee. et F. N. Wei) breeding population and extraction of core collection using cpSSR, nSSR and phenotypic markers” (Manuscript ID: forests-1805236). Those comments are all valuable and very helpful for revising and improving our paper, as well as the important guiding significance to our researches. We have studied comments carefully and have made correction which we hope meet with approval. Revised portion are marked in red in the paper. And we also upload the word file of the point-by-point response in the attachment. The main corrections in the paper and the responds to yours comments are as following:
- Dear Authors! Why didn't you write it in the conclusion?! This phenomenon has already been observed in other wild species, but the authors have proved it with their research. This is knowledge that every researcher should use in their biodiversity research.
Response to comment: We have made correction according to the yours comments. We supplied a section of conclusions in text, and added this results (only a part of the collection shows significant genetic variability, which will facilitate the breeding process, and phenotypic variability does not always go hand in hand with molecular variability) in the conclusion. Please review line 402-405.
- The authors misuse the term "germplasm". This term is very imprecise, which sometimes makes it difficult to understand the content. Sometimes it is better to use: plant, individuals, genotype, genome, DNA, cpDNA, etc. For example "„... fresh leaves from each germplasm...” substitute „... fresh leaves from each individuales/plants...”
Response to comment: We are very sorry for our negligence of the misuse the term “germplasm”. We have revised the misuse of “germplasm” in the text of manuscript. Please review line 26, 98, 108, 159, 201, 202, 205, 207, 239, 242, 282-284.
- Coloring of figures: The description of the colors should be included not only in the text but also in the legend under the figures. Figure 2 please enter brighter and more contrasting colors and in Figure 3 more contrasting. This will facilitate the analysis.
Response to comment: Considering the yours suggestion, we have supplied the description of the color in the legend under the figures. And, the colors in Figure 2 and Figure 3 replaced by brighter and more contrasting colors to facilitate the analysis in the manuscript. Please review line 231-235, 276-277, and the revised Figure 2 and Figure 3.
- Please add the table with the tested material: 1. column: name of the Provinces in China; 2. column: the designations of the test subjects (eg ZNcs1 - ZNcs ...); 3. column: the geographical coordinates of the Provinces. (Don't forget to change the order of the tables in the text!! :-)). This will make it easier for the reader to analyze Figure 1A, 3C and Table S4.
Response to comment: As you suggested that we have supplied the table with the detail of the materials. Please review line 119 (Table 1). And we have changed the order of the tables in the text. Please review line 114, 189, 210, 240, 247, 286, and 194.
- Please unify the order of the subchapters in the Methods and Results sections: Molecular research first, then phenotypic, or vice versa. Include this change in the title of your publication.
Response to comment: Considering the yours suggestion, we have unify the order of the subchapters in the parts of methods and results, and we also change the title according the unify order. The core collection were established by the data of molecular and phenotypic, so we ranked this subsection at the bottom. Please review line 115, 131, 151, 158 and the title of the manuscript.
- Please indicate more clearly the research objectives in the Introduction and the sections: Conclusions and Author Contributions.
Response to comment: As you suggested that the research objectives should be more clearly. The objectives of this study were to (1) reveal the characteristics of genetic diversity at both a molecular level and a phenotypic level, (2) extract a core collection, and then to (3) provide a foundation for effectively utilizing P. zhennan genetic resources. Please review line. And we also supplied the sections of Conclusions and Author Contributions in the text. Please reviewer line 94-98, 399-409, and 410-413.
- Chapter: Materials and Methods - DNA Isolation. "Genomic DNA extractin" is not an appropriate term. "Genomic" is usually used to refer to nuclear DNA, and you have also studied cpDNA. I suggest you write "DNA extraction"
Response to comment: We are very sorry for our negligence of the misuse in the chapter. We have made correction according to the yours suggestion. Please review line 117.
- Figure 4B. is indistinct.
Response: We have repainted Figure 4B to enhance its resolution. Please review the revised Figure 4 in line 295.
- Table S1 and S2; First column: not "Marker" but "Locus"
Response to comment: Considering the yours suggestion, we have replaced “Marker” in Table S1 and S2 by “Locus”. Please review the revised Table S1 and Table S2.
- Table S4: „Molecular ID” better: „Molecular identity". The „ID” is usually restricted to official databases.
Response to comment: Considering the yours suggestion, we have replaced “Molecular ID” by “Molecular identity” in Table S4. Please review the revised Table S4.
- The names of Chinese provinces should consistently be spelled without quotation marks.
Response to comments: As your suggested that we have revised the spell type about the Chinese province to without quotation marks. Please review line 282-284.
- Section 3.1.1, not “markers”, “marker” but “loci”, “locus”
Response to comment: Considering the yours suggestion, we have replaced “markers” and “marker” by “loci” and “locus”. Please review line 195 and 198.
- Section 3.1.1, “14 Na cross” in table 1, it is 11.
Response to comment: We are very sorry for our negligence of the mistake. We have revised it. Please review line 207.
- Section 3.4, “P. zhennan” should be Italic text.
Response to comment: We are very sorry for our negligence of the mistake. We have revised it. Please review line 239.
- “These results support the validity of the method for extracting the core collection and further suggest that the core collection effectively represents the whole germplasms” would be in discussion or conclusion.
Response to comment: As yours suggestion that we write this content in the conclusion. Please review line 405-406.
Special thanks to you for your good comments.
We appreciate for your warm work earnestly, and hope that the correction will meet with approval.
Once again, thank you very much for your comments and suggestions.

Reviewer 3 Report
The manuscript is well prepared and well written. The applied methods and the statistics are appropriate. Only minor errors and report interesting findings.
In this study, the authors explore about the genetic diversity determination of Nanmu (Phoebe zhennan S. Lee. et F. N. Wei), a subtropical evergreen broad-leaves typical of Chinese forests, through molecular and phenotypic data joint analysis to improve its breeding program. Contrary to previous works that detected low genetic diversity for four natural Nanmu population by using AFLP markers, the authors found a relatively high genetic diversity based on phenotypic traits, nSSR, and cpSSR marker analysis for 102 Nanmu germplasm.
Comments.
1) For a greater completeness of molecular characterization, I suggest introducing in the Table 1 the Number of private alleles.
2) Please write in full: “eleven” in the sentence “11 alleles (PIC = 0.303) were detected in 102 germplasms from five cpSSR loci, indicated inferior polymorphism and conservation of chlorotypes genome in P. zhennan.”
Author Response
Dear Professor:
Thank you for your letter and for the reviewers’ comments concerning our manuscript entitled “Genetic diversity of Nanmu (Phoebe zhennan S. Lee. et F. N. Wei) breeding population and extraction of core collection using cpSSR, nSSR and phenotypic markers” (Manuscript ID: forests-1805236). Those comments are all valuable and very helpful for revising and improving our paper, as well as the important guiding significance to our researches. We have studied comments carefully and have made correction which we hope meet with approval. Revised portion are marked in red in the paper. And we also upload the word file of the point-by-point response in the attachment. The main corrections in the paper and the responds to yours comments are as following:
- For a greater completeness of molecular characterization, I suggest introducing in the Table 1 the Number of private alleles.
Response to comment: Considering the yours suggestion, we have supplied the number of private alleles in Table 2 (the original Table 1). Please review the revised Table 1.
.
- Please write in full: “eleven” in the sentence “11 alleles (PIC = 0.303) were detected in 102 germplasms from five cpSSR loci, indicated inferior polymorphism and conservation of chlorotypes genome in P. zhennan.”.
Response to comment: As you suggested that we have replaced “11 alleles…” by “Eleven alleles” in this sentence. Please review line 325.
Special thanks to you for your good comments.
We appreciate for your warm work earnestly, and hope that the correction will meet with approval.
Once again, thank you very much for your comments and suggestions.
